# Effects of Shopping Rehabilitation on Older People’s Daily Activities

**DOI:** 10.3390/ijerph19010569

**Published:** 2022-01-05

**Authors:** Naoto Mouri, Ryuichi Ohta, Chiaki Sano

**Affiliations:** 1Department of Community Medicine Management, Faculty of Medicine, Shimane University, 89-1 Enya Cho, Izumo 693-8501, Japan; arhrma@gmail.com (N.M.); sanochi@med.shimane-u.ac.jp (C.S.); 2Community Care, Unnan City Hospital, 96-1 Iida, Daito-Cho, Unnan 699-1221, Japan

**Keywords:** shopping rehabilitation, nudge, older people, aged society, physical functions, cognitive functions, motor functions

## Abstract

In an aged society, the deterioration of physical and cognitive functions is prevalent. To motivate the rehabilitation of older persons, an initiative known as “shopping rehabilitation” incorporates shopping as an element of a nudge. The purpose of this study was to clarify motor function changes and cognitive functions of participants during shopping rehabilitation, through a semi-experimental study. We measured changes in the Kihon Checklist score before and after rehabilitation interventions. A paired t-test was used to analyze changes in the overall score of the basic checklist before and after the rehabilitation intervention. In December 2020, 59 participants answered the Kihon Checklist after their shopping rehabilitation intervention. During the 6-month intervention period, the number of participants with a checklist score of 8 or higher was significantly reduced after the intervention (*p* = 0.050). In the sub-analysis, the score improved significantly for the group with families (*p* = 0.050). Improvement was observed in the group living alone, but the difference was not significant (*p* = 0.428). The shopping rehabilitation intervention improved the Kihon Checklist score. Continuous observations and research are necessary to measure the long-term effects of shopping rehabilitation and the mechanisms that foster their maintenance and effects.

## 1. Introduction

In aging societies, the deterioration of physical and cognitive functions of older persons has become a paramount concern, which is related to older people’s motivation to engage in rehabilitation. Several systematic reviews of older people (aged 65+ years) report that regular exercise improves physical and cognitive functions [1,2,3]. Providing exercise rehabilitation to older people can contribute to maintaining their physical and cognitive functions, and rehabilitation may increase the ability to stay healthy [1,2,3]. However, the motivation for rehabilitation varies depending on the person [4,5,6].

As a method to overcome this issue, the concept of a nudge for human behavioral changes is becoming popular. A nudge is a strategy that changes human behavior based on behavioral science, with concepts such as “softly boosting” and “pushing lightly with elbows” [7]. Nudge-based interventions can support people’s behavioral changes by mixing favorable behaviors with their interests. For example, one measure to prevent the littering of cigarettes in London used soccer players as a nudge [8]. In London, many people enjoy soccer; therefore, ballot bins were created, which contained two transparent sections, each of which was labeled with the name of a famous soccer player. Discarding a cigarette butt into one of the sections was used to vote for the player considered best. Consequently, butts discarded on the road were reduced by approximately 26% [8].

In addition, a meta-analysis summarized the effects on physical activity of the game application “Pokémon GO”, which is a nudge-based intervention that uses gamification. The app allows users to capture characters by walking and using the GPS function of their smartphone. The motivation to capture a character is used as a nudge. The game increases walking time, distance, and the number of steps per day among its players [9]. These examples show that incorporating nudges into rehabilitation may increase motivation for older persons who are not motivated by rehabilitation alone.

Shopping can be used as a nudge for rehabilitation, which could drive rehabilitation among older people. This is known as “shopping rehabilitation”, which allows older people to enjoy shopping and naturally engage in rehabilitation [10]. In this rehabilitation process, persons who find it difficult to leave their home and require support visit a commercial facility by being picked up from their homes and engaging in rehabilitation, as exercise is added to the shopping experience using a cart dedicated to rehabilitation. This effort to incorporate shopping into rehabilitation may act as a nudge to motivate older people [10].

In rural areas, stores may be relatively far from places of residence, which can limit older people’s ability to go shopping [11,12]. Residents who cannot use their own cars due to the aging-related deterioration of their driving skills are restricted in the amount of time they have available for shopping [13,14]. Shopping rehabilitation that includes transportation for older people can be beneficial in rural contexts.

Shopping rehabilitation that includes transportation systems is expected to relieve people who cannot use transportation easily and revitalize the local economy. Shopping rehabilitation for older people and the mechanisms underlying its effects have not been clarified. Therefore, we hypothesized that shopping rehabilitation would improve the cognitive and motor functions of older persons. Clarification of such effects could drive rehabilitation in rural communities with collaboration among rural shops and transportation systems, which could additionally improve rural economies. Therefore, this study clarified the effects of shopping rehabilitation on the motor and cognitive functions of older persons.

## 2. Materials and Methods

This study used a semi-experimental method with older people who participated in a shopping rehabilitation program in Unnan City, Japan, from June to December 2020.

### 2.1. Setting

Unnan City is one of the most rural cities in Japan. It is located to the southeast of the Shimane Prefecture. In 2020, the total population of Unnan was 37,638 (18,145 males and 19,492 females), with 39% of residents aged over 65 years; this number is expected to reach 50% by 2025 [11].

### 2.2. Shopping Rehabilitation

Marcherises, where this study was conducted, is the largest commercial facility in Unnan City, and many older people visit the area daily for shopping. At Hikari Salon Unnan, an initiative known as “shopping rehabilitation” is being carried out in its commercial facility.

The participants were recruited in Unnan City. Information regarding the shopping rehabilitation was provided via brochures and local newspapers to eligible people. Those eligible for shopping rehabilitation had one or more Kihon Checklist scores available; the Japanese government constructed the checklist to detect people that required possible care [15]. The eligible people applied for the shopping rehabilitation voluntarily.

Regarding the shopping rehabilitation, 95% of the participants used shuttle cars to travel from their homes to commercial facilities. After clerks performed health checks, such as assessments of blood pressure, pulse rate, and temperature in the reception area, nine to 12 participants moved around the facility while following the instructions of two occupational therapists (OT). For this, a uniquely crafted shopping cart was used. This cart is manufactured by Hikari project Co., Ltd which is located at Unnan City, Shimane, Japan. and designed to promote effective rehabilitative movements during shopping. The Japanese government registered this cart as safe. Participants move their bodies up and down to pick up the product they want to purchase under the direction of the OT. In addition, the OT monitored participants’ postures during movements and guided them to make their exercises effective. Rehabilitation using the special shopping cart was performed for 30 min, followed by 60 min of gymnastics conducted in the Hikari Salon on the second floor which can accommodate more than 20 participants, followed by a second 30 min shopping session. Therefore, one complete session lasted approximately 120 min in total. The shopping rehabilitation was held four times a month on average. For 10 min, OTs instructed the participants regarding home exercises (Figure 1). A logic model was presented in previous studies of shopping rehabilitation [16]. It demonstrated that continuous shopping rehabilitation improved athletic performance and one’s quality of relationships, thereby reducing loneliness [15,16] (Figure 2). This research represents a pilot study that confirms the validity of the logic model through a quantitative study based on the logic model.

### 2.3. Measurements

#### 2.3.1. Primary Outcome

Changes in Kihon Checklist scores before and after shopping rehabilitation, which were shown to correlate with prognosis, were measured [15]. The Kihon Checklist was collected and compared before participation and six months after engaging in shopping rehabilitation. It was developed as a screening method to detect older people who were at high risk of requiring long-term care [17]. It consists of 25 questions grouped into seven areas: daily life, motor function, nutrition, oral function, withdrawal, cognitive function, and depressed feelings (Table 1). The prior literature shows that a score of 8 or higher is associated with reduced future independence and survival [15,17]. This study used a 24-item checklist, excluding the unscored weight and height items.

#### 2.3.2. Independent Variables

Values before and after participation in shopping rehabilitation were used as the independent variable.

#### 2.3.3. Covariate

We used a questionnaire to collect the background data of participants. The contents were age, gender (male and female), medical history, family structure (living with family or living by oneself), dependency status (0: independent, 4: highly dependent), level of dementia (0: no symptoms, 4: severely impaired), housing environment (detached house or apartment), and history of smoking and drinking. Based on the participant’s medical history, a Charlson Comorbidity Index (CCI) score was calculated for each participant to assess the severity of their medical conditions [18]. This index measures the severity of patients’ medical conditions related to the possibility of hospital admissions and mortality. The shopping rehabilitation clerk conducted the entire data collection.

### 2.4. Statistical Analysis

The necessary sample size was calculated as 34 participants, given 80% statistical power and 5% type 1 error to detect a difference of 0.1 points between pre-and post-intervention groups, with standard deviation of 0.2 in parted t-test. The participants were categorized into two groups—those with families and those living independently—to investigate the differences in demographic data related to isolation. Student’s t-test was used to assess parametric data, and the Mann–Whitney U test was performed on non-parametric data. Numerical variables were dichotomized, namely care dependency (≥1 and 0) and the dementia scale (≥1 and 0). Differences were assessed in the pre-and post-intervention proportions of total Kihon Checklist scores ≥ 8 [17]. Cases with missing data were eliminated from the analysis. Statistical significance was defined as a *p*-value < 0.05. All statistical analyses were performed using EZR (Saitama Medical Center, Jichi Medical University, Saitama, Japan), a graphical user interface for R (The R Foundation, Vienna, Austria) [19].

### 2.5. Ethical Considerations

The hospital was assured of the anonymity and confidentiality of patients’ information. Details of this study were posted on the hospital website without disclosing any details concerning the patients. To address any questions regarding this study, the contact information of the hospital representative was also listed on the website. The purpose of this study was explained to all participants, and informed consent was obtained by the clerks of shopping rehabilitation. The Clinical Ethics Committee of our institution approved this study (approval code: 20200023).

## 3. Results

### 3.1. Demographic Data

In June 2020, 72 older people participated in shopping rehabilitation and answered the Kihon Checklist. Finally, 59 participants answered the Kihon Checklist in December 2020 after the intervention. Consent was obtained from all participants during this period. The mean age of participants in this study was 86.32 years (standard deviation = 4.67), with 93.2% being female. There was no significant difference in the background data between groups living independently and with families (Table 2).

### 3.2. Change in the Kihon Checklist

During the 6-month intervention period, the number of participants with a checklist score of 8 or higher was significantly reduced after the intervention (*p* = 0.050). In the sub-analysis, the score improved significantly for the group with families (*p* = 0.050). Improvement was observed in the group living alone, but the difference was not significant (*p* = 0.428; Table 3). During this period, no apparent adverse events due to the rehabilitation were observed.

## 4. Discussion

This study shows that incorporating the element of shopping into rehabilitation can improve the Kihon Checklist score. In addition, as the proportion of persons with a total score ≥ 8 decreases, shopping rehabilitation can improve the independence period and survival rate. The element of shopping can enhance the participation of older people who do not go shopping by themselves, improve their daily activities, and reduce the risk of frailty, leading to a better quality of life.

The improvement of the Kihon Checklist score shows the effectiveness of shopping rehabilitation, depending on living situations. Shopping rehabilitation contains physical and cognitive training factors during shopping, which can contribute to improving the score. Furthermore, participants with families improved their scores significantly. This improvement might correlate with their living conditions. Older people often receive help from their families, so they may lose the opportunity to go shopping by themselves [20,21]. In rural contexts, older people do not have multiple public transportation systems [22]. To use public transportation systems, they must wait for a long time or walk to stations far from their houses; this can prevent them from shopping [23]. Therefore, their families often support their shopping or buy their utilities [24].

In contrast, in this study, a higher proportion of participants living alone had a checklist score of 8 or higher and did not improve in physical or cognitive function through the shopping rehabilitation. Since such persons lack help in their daily life, they might not collect their utilities and groceries themselves. Furthermore, isolation can promote health issues such as frailty. This can limit an individual’s ability to travel and communicate with others, which might lead to further deterioration in their physical and cognitive functions [25]. This research investigated the effect of shopping rehabilitation according to differences in living conditions. Future studies should investigate the relationship between the effect of rehabilitation and older people’s perceptions of living conditions, such as isolation and loneliness, as aged societies can be prone to such negative aspects, due to diminished mobility among the older members of the society [26,27].

The possibility of improving the independence period and survival rate through shopping rehabilitation, as shown in this research, can be critical to sustaining aged societies. Rehabilitation itself can be effective for supporting people with disabilities in medical facilities. Most rehabilitation is performed in medical facilities, but the need for physical and cognitive rehabilitation can be high in communities, even in rural areas [28,29,30]. Specifically, rural older people can lose the opportunities to meet others within their communities because they lack mobility [11,12]. Many older people must stop driving, as encouraged by the government, because of the high risk of automobile accidents [31,32].

In this research, most participants were female, which is likely related to their previous routine in a rural context. In rural regions, the woman’s role is often housework, including purchasing groceries [33]. However, as women age, they may be unable to go shopping because they lack access to shopping malls. Through shopping rehabilitation, they could begin shopping again, and thus engage in physical movement with the support of rehabilitation specialists.

In different contexts, older people often lose their routines. As with shopping rehabilitation, regaining these routines as a nudge could revitalize older people’s activities, which could help them sustain their roles in families, leading to the preservation of their physical and cognitive functions. The investigation of lost routines among older people in communities should be studied to find new ways to revitalize such persons.

Furthermore, during the COVID-19 pandemic, rural older people could not visit various destinations because of the risk and fear of infection. COVID-19 reduced the number of activities people could undertake outside of their houses, which could cause older people’s frailty to worsen [34,35]. Various recommendations regarding exercise were provided in different countries [36]. Rural people are limited regarding their activities based on the standards of urban areas [37,38,39,40]. However, when creating the standards for activity in the context of COVID-19 in each community, various activities were specified for rural areas, such as community conferences, social prescribing, and readings on improving social interactions [41,42,43]. Shopping rehabilitation may be necessary as a form of rehabilitation to improve social interactions and cognitive and physical functions. Future studies should investigate the sustainability of participants who receive qualified rehabilitations including shopping rehabilitation during the COIVD-19 pandemic [44].

### Limitations

This study had several limitations. First, as it was performed in a single shopping mall located in a Japanese rural area, the study’s setting may not be applicable to older people in all developing and developed countries. However, the results are applicable to rural areas with a lack of social resources, aged societies, and isolated older people. Future studies should investigate these constructs in other rural settings, such as remote islands or developing countries. Another limitation pertains to the sampling method. Possible confounding factors were considered in this study; however, the randomization of the sampling process could further address potential confounds. Therefore, future studies should implement randomization to overcome this limitation.

## 5. Conclusions

The shopping rehabilitation intervention improved Kihon Checklist scores. Continuous observation and research are needed to measure the long-term effects of shopping rehabilitation on participants, regarding not only improvements in physical and cognitive abilities but also loneliness, the mechanisms that foster health maintenance, and other potential impacts.

## Figures and Tables

**Figure 1 ijerph-19-00569-f001:**
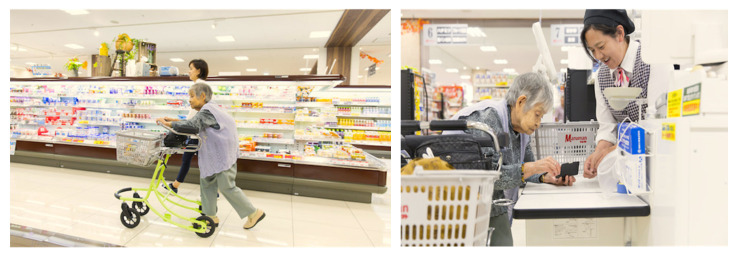
Shopping rehabilitation. Copyright © 2021 Hikari project Co., Ltd. All Rights Reserved.

**Figure 2 ijerph-19-00569-f002:**
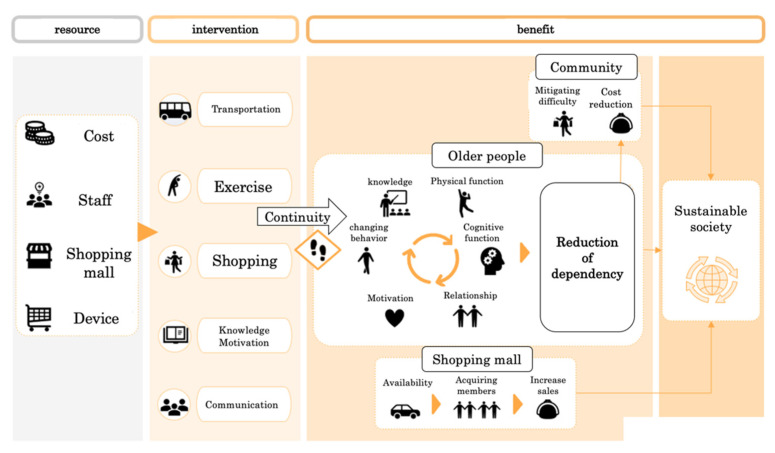
Logic model of shopping rehabilitation. Copyright © 2021 Hikari project Co., Ltd. All Rights Reserved.

**Table 1 ijerph-19-00569-t001:** Content of the Kihon Checklist.

No.	Category	Content	Answer
1	Daily life	I take a bus by myself.	yes/no
2		I buy groceries by myself.	yes/no
3		I make deposits and withdrawals myself.	yes/no
4		I visit friends’ houses.	yes/no
5		I consult with my family and friends.	yes/no
6	Motor function	I climb the stairs without railing or wall support.	yes/no
7		I get up from sitting in a chair without holding onto anything.	yes/no
8		I walk for about 15 min at a time.	yes/no
9		I have fallen in the last year.	yes/no
10		I have a lot of anxiety about falling.	yes/no
11	Nutrition	I have lost more than 2–3 kg in the past six months.	yes/no
12	Oral function	It’s more difficult for me to eat hard foods than it was six months ago.	yes/no
13		I sometimes get sick after drinking tea or soup.	yes/no
14		I’m worried about being thirsty.	yes/no
15	Withdrawal	I go out at least once a week.	yes/no
16		I go out less often than I did last year.	yes/no
17	Cognitive function	People around me say that I am forgetful, even after hearing something multiple times.	yes/no
18		I look up phone numbers and make calls by myself.	yes/no
19		Sometimes I don’t know what month or day it is.	yes/no
20	Depressed feelings	I haven’t had a sense of fulfillment in my daily life for the past two weeks.	yes/no
21		For the past two weeks, I have not been enjoying what I used to enjoy.	yes/no
22		For the past two weeks, what I used to be comfortable with now feels awkward.	yes/no
23		I don’t think I’ve been a useful person for the last two weeks.	yes/no
24		I feel tired over the last two weeks for no reason.	yes/no

**Table 2 ijerph-19-00569-t002:** Demographic data of participants.

Factor	Total	Living Independently	Living with Family	*p* Value
*n*	59	21	38	
Age	86.32 (4.67)	85.19 (5.19)	86.95 (4.31)	0.169
Females (%)	55 (93.2)	20 (95.2)	35 (92.1)	1
Dementia (%)	9 (15.3)	3 (14.3)	6 (15.8)	1
Dependency (%)	24 (40.7)	6 (28.6)	18 (47.4)	0.18
Living, detached house (%)	54 (91.5)	18 (85.7)	36 (94.7)	0.337
Smoke (%)	3 (5.8)	0 (0.0)	3 (8.6)	0.542
Alcohol (%)	8 (15.4)	2 (11.8)	6 (17.1)	1
Charlson Comorbidity Index (%)				
3	2 (3.4)	1 (4.8)	1 (2.6)	0.37
4	29 (49.2)	9 (42.9)	20 (52.6)	
5	13 (22.0)	4 (19.0)	9 (23.7)	
6	10 (16.9)	3 (14.3)	7 (18.4)	
7	4 (6.8)	3 (14.3)	1 (2.6)	
8	1 (1.7)	1 (4.8)	0 (0.0)	
Kihon Checklist score	6.71 (3.34)	6.43 (3.60)	6.87 (3.22)	0.632
Q1	0.24	0.14	0.29	
Q2	0.02	0.00	0.03	
Q3	0.14	0.10	0.16	
Q4	0.14	0.10	0.16	
Q5	0.12	0.19	0.08	
Q6	0.75	0.57	0.84	
Q7	0.39	0.38	0.39	
Q8	0.08	0.10	0.08	
Q9	0.25	0.43	0.16	
Q10	0.81	0.81	0.82	
Q11	0.15	0.14	0.16	
Q12	0.25	0.33	0.21	
Q13	0.41	0.38	0.42	
Q14	0.37	0.38	0.37	
Q15	0.02	0.00	0.03	
Q16	0.44	0.38	0.47	
Q17	0.19	0.14	0.21	
Q18	0.02	0.05	0.00	
Q19	0.32	0.48	0.24	
Q20	0.19	0.14	0.21	
Q21	0.15	0.14	0.16	
Q22	0.53	0.43	0.58	
Q23	0.29	0.19	0.34	
Q24	0.46	0.43	0.47	

**Table 3 ijerph-19-00569-t003:** Change in the number of participants with a Kihon Checklist score of 8 or higher.

	Pre-Intervention	Post-Intervention	*p*-Value
Total score ≥ 8, proportion	0.39	0.27	0.050
With family	0.37	0.24	0.050
Living alone	0.43	0.33	0.428

## Data Availability

The datasets used and/or analyzed during the current study may be obtained from the corresponding author upon reasonable request.

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
