# Peer review of "Effects of Shopping Rehabilitation on Older People’s Daily Activities"

_ijerph, 2022, doi:10.3390/ijerph19010569_

Round 1
Reviewer 1 Report
Thank you very much for the opportunity to review your paper.
Introduction: Please do not mix up exlploratory knowledge from observed society behavior like Pokemon go or smoking and intervention guide nudge strategies. This is a different thing to my point of view.
Are you sure, you study is conducted to clarify the mechanisms of shopping and their influence on function or is it rather a first approach to observe the impact.
Methods
A paired t-test is not allowed for a score, due to ordinal data.
Results
Please revise results according to revised statistics.
Discussion
Pleaser discuss how representative your "rural population shopping paradise" for Japan and generalization is.
Author Response
Responses to the Reviewers’ comments
Thank you very much for reviewing our manuscript and providing suggestions to improve on our manuscript. We have provided the point-by-point responses to the Reviewers’ comments; our revisions are in red font. Please consider our manuscript for publication.
Reviewer 1
Thank you very much for the opportunity to review your paper.
Introduction: Please do not mix up exlploratory knowledge from observed society behavior like Pokemon go or smoking and intervention guide nudge strategies. This is a different thing to my point of view.
Response:
Thank you for the positive feedback. We agree with your comment. We have added a paragraph that explains the examples in depth by describing the relationship between nudges and these activities (Lines 41 to 52).
Are you sure, you study is conducted to clarify the mechanisms of shopping and their influence on function or is it rather a first approach to observe the impact.
Response:
We agree with the reviewer’s query. We have revised the relevant paragraph comprehensively by stating a clear research question and purpose with supporting information (Lines 62 to 75).
Methods
A paired t-test is not allowed for a score, due to ordinal data.
Response:
We have revised the analysis section accordingly by using statistical methods suitable for ordinal data (Lines 147 to 159).
Results
Please revise results according to revised statistics.
Response:
We revised the results based on new analytical approach (Lines 178 to 183).
Discussion
Please discuss how representative your "rural population shopping paradise" for Japan and generalization is.
Response:
We have revised the discussion based on the suggestion, including the applicability of our research to other contexts (Lines 202 to 213, 223 to 234, and 249 to 257).
Reviewer 2 Report
The initiative of the study arouses my interest. However, some questions need to be more detailed:
2.2. Shopping rehabilitation
“After performing health checks such as vital checks” (line 83)
Does this mean that you check the health and vitality of the participants just before you start shopping? If so, where is this checkup done?
participants move around the facility while following the instructions of an occupational therapist (OT). (line 84))
How many older people go to the supermarket at the same time?
Each older person is assigned an OT
How many OTs have been needed for the program?
the OT monitors participants’ postures during movements and 89 guides them to make their exercises effective. Rehabilitation using the special shopping 90 cart is performed for 30 minutes each, with 60 minutes of gymnastics in between. One 91 session lasts around 120 minutes in total (lines 89-91)
The 60 minutes of gymnastics, where do they take place?
How many older people participate in each of these gym sessions?
30 minutes of exercises with cart + 60 minutes of gymnastics + 30 m.? (it is not explained what is done during this time) = 120 minutes
Participants are instructed on home exercises (line 93)
Who instructs them? Where are they instructed? how long?
one’s quality of relationships, thereby reducing loneliness (line 95)
On what results of the study is this claim based?
2.4. Statistical analysis
Regarding the sample size calculation, participants were needed line (136)
How was the sample size calculated?
2.5. Ethical considerations
Information related to this study was posted on the hospital website without 145 disclosing any details concerning the patients. (lines 145-146)
Who recruits the sample? How is it recruited?
Author Response
Responses to the Reviewers’ comments
Thank you very much for reviewing our manuscript and providing suggestions to improve on our manuscript. We have provided the point-by-point responses to the Reviewers’ comments; our revisions are in red font. Please consider our manuscript for publication.
Reviewer 2
The initiative of the study arouses my interest. However, some questions need to be more detailed:
2.2. Shopping rehabilitation
“After performing health checks such as vital checks” (line 83)
Does this mean that you check the health and vitality of the participants just before you start shopping? If so, where is this checkup done?
Response:
We have added when and where vital signs were assessed and what signs were checked.
participants move around the facility while following the instructions of an occupational therapist (OT). (line 84))
How many older people go to the supermarket at the same time?
Response:
We have added the numbers of participants engaging in the rehabilitation at any one time.
Each older person is assigned an OT
How many OTs have been needed for the program?
Response:
We now list the number of OTs supporting the shopping rehabilitation.
the OT monitors participants’ postures during movements and 89 guides them to make their exercises effective. Rehabilitation using the special shopping 90 cart is performed for 30 minutes each, with 60 minutes of gymnastics in between. One 91 session lasts around 120 minutes in total (lines 89-91)
The 60 minutes of gymnastics, where do they take place?
Response:
We have added where the gymnastics were performed to the method sections.
How many older people participate in each of these gym sessions?
Response:
We now list the number of participants participating in these sessions.
30 minutes of exercises with cart + 60 minutes of gymnastics + 30 m.? (it is not explained what is done during this time) = 120 minutes
Response:
We have added an explanation of the shopping rehabilitation components, including the time schedule, to the method sections.
Participants are instructed on home exercises (line 93)
Who instructs them? Where are they instructed? how long?
Response:
We have added descriptions of the home exercise to the method sections.
one’s quality of relationships, thereby reducing loneliness (line 95)
On what results of the study is this claim based?
Response:
We have adding references in support of this statement.
2.4. Statistical analysis
Regarding the sample size calculation, participants were needed line (136)
How was the sample size calculated?
Response:
We have revised the documentation to detail the sample size calculation.
2.5. Ethical considerations
Information related to this study was posted on the hospital website without 145 disclosing any details concerning the patients. (lines 145-146)
Who recruits the sample? How is it recruited?
Response:
We have revised the method section to detail the participants and how they were recruited.
Reviewer 3 Report
The manuscript entitled "Effects of Shopping Rehabilitation on Older People’s Daily Activities" describes a novel approach focused on improving motor and cognitive function in aged people. The study is based on the observation of the effects produced by "shopping rehabilitation", used as a "nudge", on Kihon checklist score (a score equal to 8 is often related to a major risk of indipendence and surviral reduction) , in participants categorized into two groups: those living in families and those living alone, to also investigate the impact of their living conditions. Shopping rehabilitation consists in the use of a special cart designed to promote rehabilitative exercises and specific movements during a shopping session of about 120 minutes. The results observed showed an improvement of the Kihon checklist score, remarkable, even though not statistically significant, in subjects not living indipendently.
In my opinion the manuscript proposed is pioneering for the rehabilitation strategy involved, well-written and the presented results are consistent with the experimental plan. My only suggestion is to strenghten, throughout the introduction section, the concept that targeted stimuli (Sirico F et al. "Effect of Video Observation and Motor Imagery on Simple Reaction Time in Cadet Pilots." J Funct Morphol Kinesiol. 2020 Dec 5;5(4):89. doi: 10.3390/jfmk5040089. PMID: 33467304; PMCID: PMC7739276) and rooled activities (Palermi, S. et al "F. Guidelines for Physical Activity—A Cross-Sectional Study to Assess Their Application in the General Population. Have We Achieved Our Goal?" Int. J. Environ. Res. Public Health 2020, 17, 3980. https://doi.org/10.3390/ijerph17113980) are effective to ameliorate global abilities and delay cognitive and physical deterioration in old people, but also in younger healthy or deseased ones.
Author Response
Responses to the Reviewers’ comments
Thank you very much for reviewing our manuscript and providing suggestions to improve on our manuscript. We have provided the point-by-point responses to the Reviewers’ comments; our revisions are in red font. Please consider our manuscript for publication.
Reviewer 3
The manuscript entitled "Effects of Shopping Rehabilitation on Older People’s Daily Activities" describes a novel approach focused on improving motor and cognitive function in aged people. The study is based on the observation of the effects produced by "shopping rehabilitation", used as a "nudge", on Kihon checklist score (a score equal to 8 is often related to a major risk of indipendence and surviral reduction) , in participants categorized into two groups: those living in families and those living alone, to also investigate the impact of their living conditions. Shopping rehabilitation consists in the use of a special cart designed to promote rehabilitative exercises and specific movements during a shopping session of about 120 minutes. The results observed showed an improvement of the Kihon checklist score, remarkable, even though not statistically significant, in subjects not living indipendently.
Response:
Thank you for the positive feedback. We have revised the manuscript based on the suggestions.
In my opinion the manuscript proposed is pioneering for the rehabilitation strategy involved, well-written and the presented results are consistent with the experimental plan. My only suggestion is to strenghten, throughout the introduction section, the concept that targeted stimuli (Sirico F et al. "Effect of Video Observation and Motor Imagery on Simple Reaction Time in Cadet Pilots." J Funct Morphol Kinesiol. 2020 Dec 5;5(4):89. doi: 10.3390/jfmk5040089. PMID: 33467304; PMCID: PMC7739276) and rooled activities (Palermi, S. et al "F. Guidelines for Physical Activity—A Cross-Sectional Study to Assess Their Application in the General Population. Have We Achieved Our Goal?" Int. J. Environ. Res. Public Health 2020, 17, 3980. https://doi.org/10.3390/ijerph17113980) are effective to ameliorate global abilities and delay cognitive and physical deterioration in old people, but also in younger healthy or deseased ones.
Response:
Thank you for the positive feedback. We have added explanations regarding stimuli such as nudge and the reference listed to strengthen our manuscript.
Round 2
Reviewer 1 Report
Thank you for addressing my comments. I am fine with the paper in its current form.